# Cessation of smoking trial in the emergency department (CoSTED): protocol for a multicentre randomised controlled trial

Caitlin Notley ,[1] Lucy Clark ,[2] Pippa Belderson,[1] Emma Ward,[1] Allan B Clark,[2] Steve Parrott,[3] Sanjay Agrawal,[4] Ben M Bloom,[5] Adrian A Boyle,[6] Geraint Morris,[7] Alasdair Gray,[8] Tim Coats,[9] Mei-See Man,[2] Linda Bauld ,[10] Richard Holland,[11] Ian Pope[1,12]

Correspondence to
Professor Caitlin Notley;
C.Notley@uea.ac.uk

## ABSTRACT

**Introduction** Attendees of emergency departments (EDs) have a higher than expected prevalence of smoking. ED attendance may be a good opportunity to prompt positive behaviour change, even for smokers not currently motivated to quit. This study aims to determine whether an opportunist smoking cessation intervention delivered in the ED can help daily smokers attending the ED quit smoking and is cost-effective.

**Methods and analysis** A two-arm pragmatic, multicentred, parallel-group, individually randomised, controlled superiority trial with an internal pilot, economic evaluation and mixed methods process evaluation. The trial will compare ED-based brief smoking cessation advice, including provision of an e-cigarette and referral to local stop smoking services (intervention) with the provision of contact details for local stop smoking services (control). Target sample size is 972, recruiting across 6 National Health Service EDs in England and Scotland. Outcomes will be collected at 1, 3 and 6 months. The primary outcome at 6 months is carbon monoxide verified continuous smoking abstinence.

**Ethics and dissemination** The trial was approved by the South Central—Oxford B Research Committee (21/SC/0288). Dissemination will include the publication of outcomes, and the process and economic evaluations in peer-reviewed journals. The findings will also be appropriately disseminated to relevant practice, policy and patient representative groups.

**Trial registration number** NCT04854616; protocol V.4.2.

## STRENGTHS AND LIMITATIONS OF THIS STUDY

⇒ This is the first study in the UK to assess the effectiveness of an emergency department (ED) delivered smoking cessation intervention using e-cigarettes opportunistically to aid smoking cessation in those not currently seeking smoking cessation advice.

⇒ The intervention development involved extensive patient and public involvement, particularly in the choice of e-cigarette.

⇒ The behavioural component of the intervention is person-centred, theory-based, scripted and delivered by advisors trained to deliver the support in the ED.

⇒ A mixed methods process evaluation will assess the delivery, implementation, fidelity and contamination of the intervention through site observations and interviews with both staff and participants. An economic evaluation will establish the cost-effectiveness of the intervention.

⇒ A potential limitation is that the control group received written information about local smoking cessation services rather than true 'usual care', which does not include written information

## INTRODUCTION

Smoking cessation interventions usually target people who demonstrate initial motivation to quit. However, evidence suggests that 'unmotivated quitters', that is, people not actively seeking smoking cessation support at that time, are also aware of the health risks of smoking and many of them are willing to consider quitting.[1] Smoking rates are four times higher in the most disadvantaged populations compared with the most affluent,[2] and those with mental health problems are far more likely to smoke.[3 4] Both groups are also more likely to attend the emergency department (ED).[5] Our patient and public involvement (PPI) work, undertaken in 2019, showed that approximately 24% of ED attendees were current smokers, many more than in the wider population estimate of 15% in that same year.[6] This PPI work was undertaken at four EDs across the UK where a cross section of attendees were asked their smoking status. An ED visit is an ideal opportunity to screen for smoking status and deliver a smoking cessation intervention. There is good evidence that smokers in the ED are willing to consider quitting.[7]

Previous public health interventions in the ED have failed due to time pressures on ED staff delivering those interventions.[8]

There is inconsistent evidence for smoking cessation interventions delivered in the ED,[9] no UK evidence, and no studies have explored using e-cigarettes for smoking cessation support within the ED. E-cigarettes are now the most popular support method for smokers trying to stop[6] and there is increasing evidence that they are an effective way to quit.[10] A Cochrane living review suggests e-cigarettes are two times as effective as nicotine replacement therapy when combined with behavioural support.[11] However, initial start-up costs and lack of information may be barriers to first use.[12] Providing an e-cigarette starter kit and brief advice, plus referral to support from local stop smoking services, may be a highly effective way of helping people to stop smoking, but has so far not been tested in populations recruited in an opportunistic setting that include unmotivated quitters and people who were not planning to access smoking cessation support.

This research will provide much needed evidence as to whether a smoking cessation intervention delivered in EDs using e-cigarettes is effective and improves treatment uptake in high prevalence disadvantaged population groups. This is key to meeting Government targets for populations in greatest need, including those in lower socioeconomic and hard to reach groups, and reducing health inequalities.[13 14] This study will also provide evidence of the effectiveness of e-cigarettes for smoking cessation in smokers not planning a quit attempt.

The overarching aim of this trial is to determine if an individually tailored brief intervention that includes behavioural support, provision of an e-cigarette and referral to local smoking cessation services, provides an effective and cost-effective approach to supporting smoking abstinence and/or reduction in unmotivated quitters attending the ED.

The specific aims of the trial are to:
▶ Run an internal pilot, with clear stop/go criteria, primarily to test recruitment systems.
▶ Definitively test real-world effectiveness of an ED-based brief, tailored, smoking cessation intervention in comparison with usual care, by comparing smoking abstinence at 6 month follow-up between-trial groups.
▶ Undertake a cost-effectiveness analysis of the intervention in comparison with enhanced usual care from a National Health Service (NHS) and personal social services (PSS) perspective.
▶ Undertake an embedded mixed methods process evaluation to assess intervention delivery, implementation, fidelity and contamination.

This protocol is reported in accordance with the Standard Protocol Items: Recommendations for Interventional Trials recommendations[15] and the Template for Intervention Description and Replication guidelines for intervention description.[16]

## Study design and setting

The Cessation of smoking trial in the emergency department (CoSTED) trial is a pragmatic two-group, multi-centred, randomised, controlled superiority trial with an internal pilot, economic evaluation and process evaluation. It will compare ED-based brief, individually tailored, smoking cessation advice including provision of an e-cigarette and referral to local stop smoking services (intervention) with the provision of contact details for local stop smoking services (control).

Recruitment to the trial will be over approximately 9 months from 4 January 2022 in six NHS EDs preselected to maximise generalisability of findings based on being representative in deprivation, ethnicity, hospital size (including teaching and district general hospitals) and location (urban and rural). Additional sites will be included if recruitment to target is challenging.

## Patient and public involvement

One member of the public is involved as named member of the trial steering group and a further three PPI representatives are involved in the trial management group (TMG) and advisory group. Three PPI members contributed fully to intervention development by commenting, rating and assessing different e-cigarette starter kits, to enable informed choice of the intervention starter kit for the trial. PPI members have actively contributed to study design and materials and will continue to be consulted throughout the study and invited to contribute to dissemination.

## Study population

Potential participants are approached in the ED. Participants are adults (≥18 years) who are daily tobacco smokers (≥1 cigarette or equivalent per day), verified by submitting an expired carbon monoxide (CO) breath test reading of ≥8 ppm. Smokers are ineligible if they require immediate medical treatment as defined by the treating clinician, are in police custody, report a known history of allergy to nicotine replacement products, are already using an e-cigarette daily or do not have capacity to give informed consent for participation. For those requiring translation, the method already used for triage and clinical appointments in the ED is used, following local policies.

Potential participants admitted to hospital are not excluded, and if allocated to intervention, still receive it in the ED. They are given information on the local hospital smoking and e-cigarette policies.

## Study procedures
### Participant identification, approach and consent

Potentially eligible participants are identified during their ED attendance by an ED researcher, or medical or nursing staff. Patients reporting smoking tobacco are provided with an information sheet (see figure 1). If eligible and interested, participants provide informed consent during their ED visit.

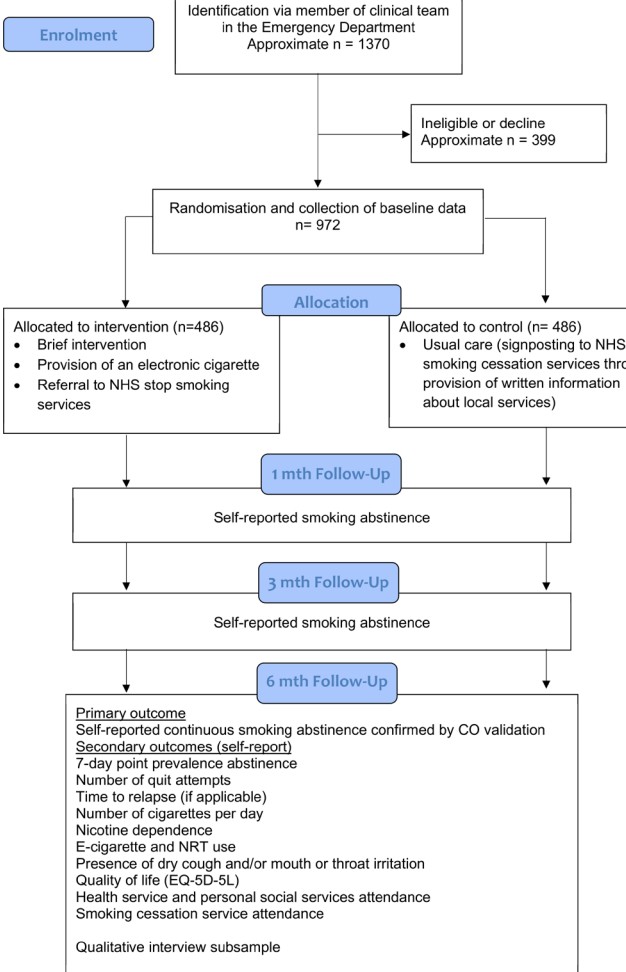

**Figure 1** Participant flow diagram. CO, carbon monoxide; NHS, National Health Service; NRT, Nicotine replacement therapy; EQ-5D-5L, quality of life validated measure

In the event of a person accompanying an eligible patient also meeting inclusion criteria and requesting to participate, the following procedure is followed: (1) if the patient did not consent to participate but the person accompanying them did consent, the person accompanying the patient is randomised and (2) if both the patient and the person accompanying them consented to participate, the patient is randomised and the accompanying person is allocated to receive the same treatment (intervention or control), but is not included as part of the trial population for the primary analysis.

### Baseline assessment

Participants complete a baseline questionnaire face to face with a member of the local study team (table 1).

### Randomisation

Participants are individually randomised to intervention or control (1:1 ratio). Randomisation is by automated web-based system, computer generated by the United Kingdom Clinical Research Collaboration (UKCRC)-registered Norwich Clinical Trials Unit (NCTU) using a blocked design, stratified by site.

### Blinding

Due to the participatory nature of the intervention, it is impossible to blind participants or treating advisors to group allocation. Members of the trial management team will not be blinded to group allocation when assessing summary statistics for monitoring purposes throughout the trial. Analysis of the primary and secondary outcomes will be undertaken by a statistician blind to group allocation.

### Internal pilot

The Independent Data Monitoring Committee (DMC) and Independent Trial Steering Committee (TSC) will scrutinise recruitment and protocol fidelity at 3 months into recruitment to establish continuation or stopping the trial at the pilot stage.

### Follow-up
#### 1-month and 3-month follow-up

At 1 and 3 months post randomisation, text messages are automatically generated by the study database and sent to all trial participants to ask about smoking status and any hospital admissions (to capture potential serious adverse events). If after two attempts there is no response, the central trial team follow-up by email, telephone, text or post.

#### 6-month follow-up

At 6 months post randomisation, a link is automatically generated by the database and sent by text/email (depending on participant preference). The questionnaires consist of smoking status, time to relapse (if applicable), nicotine product use, e-cigarette use, use of healthcare and smoking cessation services in the past 6 months, self-reported dry cough and throat irritation, the Fagerstrom test for nicotine dependence,[17] the Motivation to Stop Smoking Scale,[18] quality of life (EQ-5D-5L)[19] and health service use (which captures potential serious adverse events). If participants do not respond the central trial team follows-up by email, telephone, text or post.

For those who report smoking abstinence at 6 months, biochemical validation is requested by CO breath test by three potential mechanisms. Participants can either (1) return to the hospital they attended at baseline, (2) undertake remote collection with a CO monitor that is sent to them (an appointment will be made with the central or local team for a video call to retrieve the reading), (3) be visited at their home by a member of the local or central research team or (4) meet with a member of the local or central team at a mutually acceptable place. On completing the 6-month follow-up questionnaire, randomised participants are reimbursed for their time with a £30 shopping voucher. Those who report no longer smoking and also complete the CO verification are offered an additional £30 voucher as reimbursement for their additional time, irrespective of the value of the CO reading.

**Table 1** Schedule of enrolment, assessments and interventions

| Timepoint | Enrolment | Allocation | Post allocation | | |
| --- | --- | --- | --- | --- | --- |
| | At emergency department attendance | Following eligibility | Following allocation | 1 and 3 months of follow-up | 6-month follow-up |
| Enrolment | | | | | |
| Eligibility screen | X | | | | |
| Informed consent | X | | | | |
| Check eligibility including carbon monoxide monitor reading | X | | | | X |
| Allocation | | X | | | |
| Assessments | | | | | |
| Demographics | | X | | | |
| Self-reported smoking status | | X | | X | X | X |
| Quit attempts | | X | | | X |
| Time to relapse | | | | | X |
| Cigarette and tobacco use | | X | | | X |
| Nicotine dependence[17] | | X | | | X |
| Motivation to stop smoking[33] | | X | | | X |
| E-cigarette use | | X | | | X |
| Incidence of dry cough | | X | | | X |
| Incidence of mouth or throat irritation | | X | | | X |
| Quality of life—EQ-5D-5L[19] | | X | | | X |
| Self-reported use of healthcare services | | X | | | X |
| Adverse events | | | | X | X | X |
| Self-reported use of smoking cessation services | | X | | | X |
| Interventions | | | | | |
| Smoking cessation intervention | | | X | | |
| Referral to smoking cessation service | | | X | | |
| E-cigarette given | | | X | | |
| Signposting to National Health Service smoking cessation services | | | X | | |

## Trial allocation groups

### Control

All trial participants are offered signposting to local NHS stop smoking services through provision of written information. In the control, arm this information is provided to the participant by the member of the research team taking consent; for the intervention group it forms part of the written information provided as part of the intervention.

### Intervention

Those randomised to the intervention receive the CoSTED intervention, which is an opportunistic smoking cessation intervention undertaken in the ED, comprising three elements: (1) brief theory-driven[20] smoking cessation advice (up to 15 min), (2) the provision of an e-cigarette plus instruction in its use (up to 15 min) and (3) referral to local stop smoking services.

Following trial specific intervention training, the advice will be delivered individually (or with an accompanying person) by a smoking cessation advisor. Protocol driven advice addresses key aspects of the importance of switching away from tobacco smoking, tailored to the participant (eg, discussing improved wound healing for patients attending with an open wound or discussing

increased risk of fractures for those who continue to smoke, for those attending with a fracture). This part of the intervention is a single session undertaken within the ED department, in a quiet area or in a separate room if available. Participants are provided with an e-cigarette starter kit (the DotPro starter kit, manufactured by Liberty Flights, an independent e-cigarette manufacturer), which includes approximately 2 weeks supply of refill pods and training on its use. Participants are also offered referral to the local stop smoking service who will provide routine follow-up.

Those delivering the intervention, mainly nurses and healthcare assistants, undertook 2.5 days of standardised training and were provided with an intervention manual and videos. Training included online modules from the National Centre for Smoking Cessation (NCSCT) (NCSCT Training and Assessment Programme—core components, then specialty courses for certified practitioners, including modules on 'mental health and smoking cessation', 'pregnancy and smoking cessation', 'E-cigarettes: a guide for healthcare professionals' and 'very brief advice on secondhand smoke'). Following this, NCSCT level 2 smoking cessation advisor training was delivered by SmokeFree Norfolk and bespoke training on use of e-cigarettes, which was initially delivered face to face and then videorecorded for future training. All those trained had the opportunity to undertake some role-play and/or shadow a trained advisor prior to delivering the intervention themselves. Methods to enhance and assess intervention fidelity are informed by the National Institutes of Health Behaviour Change Consortium fidelity framework.[21]

### Sample size
The sample size was based on an estimated quit rate of 12.2% in the intervention group. This was based on a US trial of an ED smoking cessation intervention using a brief intervention, referral to smoking cessation services and nicotine replacement.[22] A quit rate of 6.2% was used in the control group based on an average of three studies of unmotivated quitters who received either signposting or no intervention.[23–25] This is also similar to the results of a recent Cochrane review of incentives reporting a quit rate of 7.2% among controls.[26] A sample size using 90% power gives a required sample size of 972 (486 per group) at the 5% level of significance using a two-tailed test. No increase for dropout has been made as dropouts will be assumed to have returned to smoking as is standard in the smoking cessation literature.[27]

### Outcome measures
See table 1.

### Primary outcome
The primary effectiveness outcome is self-reported continuous smoking abstinence, biochemically validated by CO monitoring at 6 months with a cut-off of ≥8 ppm (ie, a reading of ≤7 will denote abstinence),

according to the Russell standard.[27] If CO readings cannot be gathered, the participant is assumed to be smoking.

### Other outcome measures
Secondary outcomes measured at 6 months from randomisation:
1. 7-day point prevalence abstinence (ie, current smoking status, self-report of having smoked no cigarettes (not even a puff) in the past 7 days, biochemically validated by CO monitoring with cut-off of ≥8 ppm).
2. Number of quit attempts.
3. Time to relapse (if applicable).
4. Number of cigarettes per day.
5. Nicotine dependence.[17]
6. Number of times using an e-cigarette per day.
7. Incidence of self-reported dry cough or mouth or throat irritation.[10]
8. Motivation to stop smoking.[25]
9. Self-reported use of healthcare services in the last 6 months.
10. Self-reported use of smoking cessation services in last 6 months.
11. Quality of life (using the EQ-5D-5L).[19]
12. Self-reported smoking status and adverse events/reactions will be collected at 1 and 3 months.

### Economic evaluation
The economic evaluation takes the form of an incremental cost-effectiveness analysis (CEA) from an NHS/PSS perspective on an 'intention to treat' (ITT) basis following National Institute for Clinical Excellence (NICE) guidance.[28] The costs comprise intervention/usual care costs and wider healthcare service costs. Intervention costs are prospectively recorded alongside the intervention delivery. A self-completion service use questionnaire is used to collect participants' utilisation of wider healthcare services at baseline and 6-month follow-up. A set of national average unit costs from secondary sources will be applied to derive a healthcare cost profile for each patient. We also collect participants' out-of-pocket expenditure in relation to smoking cessation to capture potential financial burdens for participants.

The economic evaluation will use both smoking abstinence at 6 months post randomisation and quality-adjusted life years, derived from EQ-5D-5L,[19] as outcomes.

The primary economic analysis is a within-trial CEA using total costs over the trial period and smoking abstinence at 6-month follow-up. As the effects of smoking cessation on health and healthcare utilisation are likely to occur beyond the trial period, to demonstrate the effects beyond the time horizon of the trial we will undertake model-based lifetime projections to estimate the long-term cost-effectiveness of the intervention. To reflect the full impact of smoking cessation on health, we will also estimate lifetime incremental cost-effectiveness ratios using model-based projections.

## Process evaluation

A mixed methods process evaluation[26] will assess implementation and explore participant views on the intervention compared with usual care, contextual variation and potential contamination between the intervention and control group.

### Participant interviews

Detailed qualitative data will be collected through semi-structured interviews with a purposive sample of both intervention and control group participants (total n=30) after completion of final follow-up. Participants will have been given the option at baseline to consent to being contacted about the qualitative interviews and formal consent will be sought.

Interviews will be undertaken face to face or remotely via telephone or video-conferencing and audio-recorded for transcription. The topic guide for intervention participants will enquire about views and experiences of the intervention, barriers and facilitators, and explore their longer-term perspective beyond the intervention period. For control participants, the topic guide will explore experience of trial participation and access to usual care smoking cessation support, or any other form of support accessed.

### Smoking cessation advisor interviews

At least one smoking cessation advisor from each of the six ED sites delivering the intervention will take part in a qualitative interview on completion of recruitment. Staff interviews will assess views and experiences of intervention delivery, giving an insider perspective on which parts of the intervention package are deemed to be most helpful, in which circumstances and with which participants. Barriers and facilitators to intervention delivery will be explored from the staff perspective, to aid interpretation of trial findings and triangulate with participants' qualitative data.

Interview data will be analysed thematically,[29] using a combination of inductive and deductive approaches to assess process elements of the trial. Initially, all data will be analysed deductively, guided by the Medical Research Council (MRC) guidance for complex interventions.[30 31] Data will then be analysed inductively and more broadly. This will include critiquing the conceptual approach of CoSTED, guided by its logic model understanding any unintended consequences and reflections on the intervention from the participants' perspectives.

### Site observations

Observational data of the intervention delivery and ED setting will be collected at each of the six ED sites. A researcher will attend for at least 3 hours and take notes about the context, the interactions and conversation between patients approached for the trial and staff involved. A structured observation record sheet will be used for this purpose, prompting the researcher to observe the environment, interactions between key actors, the culture of the department, the flow of patients and any critical events that may impact on intervention delivery and implementation. The researcher will also broadly document details of the setting, including any smoking and vaping observed in and around the department. This will aid in assessing the context of intervention delivery. A Site Profile Form, completed prior to site observations as part of the Site Initiation Visit, will capture information that will support the researcher observations. Observation notes will be coded thematically using QSR NVivo and triangulated with interview data.

### Descriptive quantitative measures: fidelity, dose and reach

Quantitative process measures such as recruitment and consent rates, retention and drop out, will be analysed to help provide an explanatory context for eventual trial outcomes. The smoking cessation advisor will complete a brief feedback form after each intervention contact to enable monitoring of fidelity (eg, incorporation of motivational advice and device demonstration) and 'dose' (eg, duration of consultation, interruptions to intervention delivery). The reach of the intervention will be evaluated using sociodemographic data collected for all participants.

### Trial data handling

Data are collected and maintained in accordance with the current legal and regulatory requirements.

The REDCap database and associated code have been developed by NCTU Data Management, in conjunction with the CoSTED trial team. The database software provides a number of features to help maintain data quality, including maintaining an audit trail, allowing custom validations on all data, allowing users to raise data query requests and search facilities to identify validation failure/missing data.

After completion of the trial, the database will be retained on the servers of NCTU for on-going analysis of secondary outcomes and held for 5 years in line with our data retention policy.

The identification, screening and enrolment logs, linking participant identifiable data to the pseudoanonymised patient identification number (PIN), will be held centrally in REDCap to enable us to monitor screening progress and send out the SMS messages and emails. Screening forms will either be held in written form in a locked filing cabinet or electronically in a password protected form on hospital computers. After completion of the trial, the identification, screening and enrolment logs will be stored securely by the sites for 5 years.

Qualitative interview data will be collected on an audio recorder, or by using computer recording via video conference, and immediately transferred to secure cloud storage (NCTU sharepoint) with access restricted to nominated members of the research team. Audio files will be transcribed verbatim by a member of University of East Anglia (UEA) administrative staff or by an external transcription company. When transcribed, a researcher

will assign each transcript a unique study specific number and/or code and remove all identifying information from the transcripts. The anonymised data will be used for analysis and stored securely for 5 years unless otherwise advised by NCTU.

Observational notes will also be assigned a unique identifying code and stored securely at UEA with access restricted to nominated members of the research team.

## Data analysis

A full statistical analysis plan was drafted during the trial delivery phase and will be reviewed and signed off by the independent DMC and TSC prior to database lock. The analyses will be reported in full and in accordance with the Consolidated Standards of Reporting Trials guidelines.[32] The main planned analyses are summarised below.

The comparison of outcomes between the treatment groups will be based on the ITT population. This will consist of all randomised individuals according to the group that they were allocated. For individuals who have missing data for the primary outcome, it will be assumed that they are smoking. The primary outcome measure, smoking abstinence, will be compared between treatment groups using a binomial regression model adjusting for the factors stratified in the randomisation and any baseline factors that will be prespecified by the TMG prior to database lock, and the decision will be made blind to treatment allocation. The effect size will be estimated as the relative risk (log link function in the binomial regression) and the risk difference (identity link function).

Secondary outcome measures will be analysed in a similar fashion, but the regression model will depend on the outcome, specifically, binomial regression for binary outcomes, Poisson/negative binomial for count outcomes and linear regression for continuous outcomes. If appropriate, non-parametric methods will be used.

The pattern of missing data will be assessed, and multiple imputation methods will be used to assess for sensitivity of the results to assumptions around the missing data.

## DISCUSSION

This pragmatic randomised controlled trial aims to determine the effectiveness of an opportunistic smoking cessation intervention (provision of an e-cigarette, brief advice and referral for stop smoking support) delivered to participants while in an ED. This trial is unique in that smoking cessation interventions in UK EDs have never, to our knowledge, been tested in a trial. This trial will provide valuable insights into a novel location for prompting positive behaviour change.

## Current study status

The CoSTED study started participant recruitment in January 2022. Data collection for the 6-month follow-ups is expected to be completed in April 2023 and results are expected to be published in late 2023.

## ETHICS AND DISSEMINATION

The trial was approved by the South Central—Oxford B Research Committee (21/SC/0288). Dissemination will include the publication of outcomes, and the process and economic evaluations in peer-reviewed journals. The findings will also be appropriately disseminated to relevant practice, policy and patient representative groups.

**Author affiliations**
[1]Faculty of Medicine and Health Sciences, Norwich Medical School, University of East Anglia, Norwich, UK
[2]Norwich Clincial Trials Unit, Faculty of Medicine and Health Sciences, University of East Anglia, Norwich, UK
[3]Department of Health Sciences, University of York, York, UK
[4]Institute of Lung Health, University of Leicester, Leicester, UK
[5]Emergency Department, Barts Health NHS Trust, London, UK
[6]Emergency Medicine, Addenbrooke's Hospital, Cambridge, UK
[7]Department of Emergency Medicine, Homerton University Hospital NHS Foundation Trust, London, UK
[8]Department of Emergency Medicine, Royal Infirmary of Edinburgh, Edinburgh, UK
[9]Department of Cardiovascular Sciences, University of Leicester, Leicester, UK
[10]Usher Institute of Population Health Sciences and Informatics, University of Edinburgh Division of Medical and Radiological Sciences, Edinburgh, UK
[11]Medical School, University of Leicester, Leicester, UK
[12]Department of Emergency Medicine, Norfolk and Norwich University Hospitals NHS Foundation Trust, Norwich, UK

**Acknowledgements** We appreciate the support of the study sponsor, Norfolk and Norwich University Hospital NHS Foundation Trust (Norfolk and Norwich University Hospital Colney Lane Norwich NR4 7UY Tel: 01603 286286). The study was led by researchers at the University of East Anglia (UEA) and managed by the Norwich Clinical Trials Unit (NCTU) at UEA. Our thanks go to all organisations involved in recruitment and in supporting delivery of the intervention, particularly local stop smoking services. We would also like to thank PPI contributors who were critical in defining choice of e cigarette starter kit and shaping the intervention. We acknowledge the contribution of the researchers at sites who underwent training to deliver the intervention. We thank SmokeFree Norfolk for supporting the study by training all smoking advisors. Thanks also to our independent Trial Steering Committee and Data Monitoring committee members.

**Contributors** IP and CN conceived the idea for the study. CN, IP, EW and PB developed the intervention. LC is the trial manager and M-SM is the senior trial manager. AC is the study statistician, SP is the health economist. SA, BB, AAB, GM, AG and TC are site Principal Investigators. LB and RH provide academic expertise. All authors contributed to protocol writing and approved the final version.

**Funding** This study is funded by the National Institute for Health Research (NIHR) HTA programme Tobacco Cessation, Control and Harm Reduction (project reference 129438).

**Competing interests** None declared.

**Patient and public involvement** Patients and the public were involved in the design, conduct, reporting and dissemination plans of this research. Refer to the patient and public involvement section for further details.

**Patient consent for publication** Not applicable.

**Provenance and peer review** Not commissioned; externally peer reviewed.

**ORCID iDs**
Caitlin Notley http://orcid.org/0000-0003-0876-3304
Lucy Clark http://orcid.org/0000-0001-7162-0512
Linda Bauld http://orcid.org/0000-0001-7411-4260

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
