## [Reviewer comments · BMJ Open]

ARTICLE DETAILS

TITLE (PROVISIONAL)	Cessation of Smoking Trial in the Emergency Department (CoSTED): protocol for a multicentre randomised controlled trial
AUTHORS	Notley, Caitlin; Clark, Lucy; Belderson, P; Ward, Emma; Clark, Allan; Parrott, Steve; Agrawal, Sanjay; Bloom, Ben; Boyle, Adrian; Morris, Geraint; Gray, Alasdair; Coats, Tim; Man, Mei-See; Bauld, Linda; Holland, Richard; Pope, Ian

VERSION 1 – REVIEW

REVIEWER	Rasmussen, Mette Bispebjerg Hospital, Who-cc
REVIEW RETURNED	12-Jul-2022

GENERAL COMMENTS	This study protocol is very interesting and relevant, and it is well-written. I only have a few comment - and I consider my comments to be in the "proofreading" category. In general: 1) Please make sure abbreviations are explained when they are first used (e.g. ITT on page 10 line 26, ICER also page 10 line 45, UEA on page 12 line 31, or TMG on page 12 line 54).2) Make sure to use acronyms consistently throughout the text (e.g. COSTED vs. CoSTED and redcam vs. REDCap).3) Table 1. Is is not possible for me to see more than one line of text in each row! Please make sure the text is available to future readers.4) Please be consistent in the way you use e.g. "1 month" or "3-month" follow-up (e.g. see heading on page 8, line 18).5) Please be consistent with the use of capital letters in the headings.6) Page 9, line 14: Please change "e cigarette" to "e-cigarette", and replace "." after "Manufacturer)" with a ",".7) Page 9, line 51- : Please adjust the margins of the text.8) Insert space in "guidelines(29)" (page 12, line 45).9) Figure 1: I find it a bit strange that you write approximately 399! Why not approximately 400?10) Figure 1: Under Primary outcome, I suggest you include "continuous" to make it clear how it differs from the Secondary outcome of 7-day point prevalence. I wish you the best of luck with your study, and look forward to reading the results in a year or two.
---

REVIEWER	Steed, Liz Barts and The London School of Medicine and Dentistry, Centre for Primary Care and Public Health
-----------------	--

GENERAL COMMENTS

This is a well designed study and has potential to produce important outcomes. The manuscript is well written however I have highlighted a few points below which I believe would add greater clarity for the reader.

Introduction

1. In the first paragraph I am confused by the PPI work, more information to contextualise what was done would be helpful e.g. how many people did you talk to, was this an audit of some kind?

2. There is a statement that previous public health interventions haven't worked due to time. Some expansion on this discussion e.g. what sort of intervention might be suitable and how this study is addressing that may be helpful. Reference to other studies of relevance in ED e.g. reference 19 might also be usefully introduced here to contextualise the current research.

3. It would be useful to describe a little about the behavioural support that has been shown to increase efficacy of e-cigarettes

Methods

4. Please clarify the procedure for deeming patients eligible – is this at triage, where they may be asked their smoking status as routine or is it following initial review by a clinician for example

5. There is reference to 'unmotivated' smokers at several points but inclusion does not seem to assess motivation. Also 'unmotivated' suggests an active state and I wonder if you are actually referring to more of a passive state i.e. people might not be unmotivated just might not be thinking about it. I wonder if what you are actually talking about is including all people eligible whether expressing active motivation to stop or not. Some clarification around this would be helpful. For example in the introduction when you introduce the term you might clarify what you mean but 'unmotivated'

6. Please provide further clarification on why if a person attends with a 'patient' presenting to ED and both agree to participate and both get the treatment they will not be part of analysis, although they will if the 'patient' doesn't agree to participate – is there a rationale for this

7. Given randomised is individual but presumably there is significant risk of contamination particularly for the behavioural element of the intervention can the authors explain any actions taken to minimise or control for this

8. Blinding – can the authors clarify whether there will be blinding for the individuals conducting outcome analysis

9. I'm not quite clear if the training is specific to this study or standardised for all stop smoking advisors. Also are these advisors part of the routine ED team or essentially brought in as part of the research study this has implications for roll-out and implementation.

10. I think the behavioural elements of the intervention could be better specified. You mention it is theory driven what theory is this. Also has the intervention been feasibility tested. Reference to this and any related publications would be helpful.

	Primary outcome I am a little unclear whether it is self-report that is primary outcome or CO reading. What will happen if self-report not support by CO reading? References These seem to be have gone a little awry from 17 onwards e.g. 19 is not the appropriate fidelity reference Table one Check wording it currently says 'eligibility screening including' on third row but then doesn't include inclusion.
--	--

VERSION 1 – AUTHOR RESPONSE

Reviewer: 1

Dr. Mette Rasmussen, Bispebjerg Hospital, Lund University

Comments to the Author:

This study protocol is very interesting and relevant, and it is well-written.

I only have a few comment - and I consider my comments to be in the "proofreading" category.

In general:

- 1) Please make sure abbreviations are explained when they are first used (e.g. ITT on page 10 line 26, ICER also page 10 line 45, UEA on page 12 line 31, or TMG on page 12 line 54).

These have all been written in full

- 2) Make sure to use acronyms consistently throughout the text (e.g. COSTED vs. CoSTED and redcam vs. REDCap).

Thank you. These have all been checked and amended as necessary.

- 3) Table 1. It is not possible for me to see more than one line of text in each row! Please make sure the text is available to future readers.

We apologise for this. Perhaps it is the nature of the browser or programme used for review? We include table 1 in full within the manuscript text.

- 4) Please be consistent in the way you use e.g. "1 month" or "3-month" follow-up (e.g. see heading on page 8, line 18).

Amended

- 5) Please be consistent with the use of capital letters in the headings.

Checked and amended where necessary

- 6) Page 9, line 14: Please change "e cigarette" to "e-cigarette", and replace "." after "Manufacturer)" with a ",",.

Addressed

- 7) Page 9, line 51- : Please adjust the margins of the text.

The text is not formatted or typeset since this will be completed at publication stage.

- 8) Insert space in "guidelines(29)" (page 12, line 45).

Addressed

- 9) Figure 1: I find it a bit strange that you write approximately 399! Why not approximately 400?

These figures are actual figures based on screening logs received from sites. We use 'approximate' as cannot guarantee that those figures are objectively fully accurate.

- 10) Figure 1: Under Primary outcome, I suggest you include "continuous" to make it clear how it differs from the Secondary outcome of 7-day point prevalence.

This change has been made as suggested

Reviewer: 2

Comments to the Author:

This is a well designed study and has potential to produce important outcomes. The manuscript is well written however I have highlighted a few points below which I believe would add greater clarity for the reader.

Introduction

1. In the first paragraph I am confused by the PPI work, more information to contextualise what was done would be helpful e.g. how many people did you talk to, was this an audit of some kind?

Thank you for this comment. Details have been added. This was not an audit, but background 'patient and public' engagement to help inform our study development.

2. There is a statement that previous public health interventions haven't worked due to time. Some expansion on this discussion e.g. what sort of intervention might be suitable and how this study is addressing that may be helpful. Reference to other studies of relevance in ED e.g. reference 19 might also be usefully introduced here to contextualise the current research.

Thank you for this comment. We are minded to keep the introduction very short due to word limitations of the journal. We have therefore not added additional text as feel that this paragraph introduces the approach of providing dedicated (funded) stop smoking advisors as being essential to ensure this trial can recruit.

3. It would be useful to describe a little about the behavioural support that has been shown to increase efficacy of e-cigarettes

As above, we have not added additional narrative due to word length considerations. However reference (12) does provide evidence that a lack of information about how to use an e cigarette is a barrier to use, implying that increased information may help improve effectiveness of the e cigarette offer.

Methods

4. Please clarify the procedure for deeming patients eligible – is this at triage, where they may be asked their smoking status as routine or is it following initial review by a clinician for example

We have clarified under 'potential participants that 'Potentially eligible participants are identified during their ED attendance by an ED researcher, or medical or nursing staff.' Taking a pragmatic approach, this was usually following triage, during waiting time prior to initial review by a clinician. However, if patients returned to the waiting area following initial review they may have potentially been recruited at this point.

5. There is reference to 'unmotivated' smokers at several points but inclusion does not seem to assess motivation. Also 'unmotivated' suggests an active state and I wonder if you are actually referring to more of a passive state i.e. people might not be unmotivated just might not be thinking about it. I wonder if what you are actually talking about is including all people eligible whether expressing active motivation to stop or not. Some clarification around this would be helpful. For example in the introduction when you introduce the term you might clarify what you mean but 'unmotivated'

Thank you for raising this point. By 'unmotivated' in this context we imply a passive state. Participants are recruited at the point of attending the ED for whatever reason. They may of

course also be actively motivated to quit smoking, or may not be considering this at all. This distinguishes the population from what we would call 'motivated' quitters who might be attending a stop smoking service actively seeking support to quit. This trial is important as most smoking cessation trials recruit 'motivated quitters', actively seeking support to quit. In this study we recruit 'unmotivated' smokers not actively seeking support. We have clarified this as suggested to read: 'i.e people now actively seeking smoking cessation support at that time'

6. Please provide further clarification on why if a person attends with a 'patient' presenting to ED and both agree to participate and both get the treatment they will not be part of analysis, although they will if the 'patient' doesn't agree to participate – is there a rationale for this

The rationale for this is that the trial is individually randomised. With an accompanying 'other' the design got very complex thinking about dyads receiving the intervention, so the decision was taken with the trial statisticians at the initial funding application stage to not exclude accompanying 'others' but to not include them in the primary analysis. This was partly due to the uncertainty of powering the study when the number of dyads was unknown and the intraclass correlation coefficient within the dyad was also unknown. A further consideration at the design stage was that this study initiated during strict covid regulations, and thus the number of 'accompanying others' was extremely limited. Most EDs participating for most of the recruitment period did not allow an accompanying person into the ED, therefore with the low numbers of accompanying people we felt it would be 'cleaner' to exclude them from the primary analysis. They will be included within the analysis as a secondary analysis.

7. Given randomised is individual but presumably there is significant risk of contamination particularly for the behavioural element of the intervention can the authors explain any actions taken to minimise or control for this

We delivered bespoke training to the smoking cessation advisors that included not only the behavioural intervention training, but also training on the research design, emphasising the trial design and the importance of fidelity to protocol to avoid contamination. This is something that we will carefully assess through our process evaluation, as detailed in the paper. This intervention is not available online as it was developed for this study, which minimises the risk of contamination.

8. Blinding – can the authors clarify whether there will be blinding for the individuals conducting outcome analysis

Yes – the statistician undertaking the analysis will be blind to allocation. This is covered in the analysis section but has been clarified under the 'binding' sub-heading.

9. I'm not quite clear if the training is specific to this study or standardised for all stop smoking advisors. Also are these advisors part of the routine ED team or essentially brought in as part of the research study this has implications for roll-out and implementation.

Training was specific to this study (stated in the paper as 'trial specific training'), incorporating aspects of the standardised level 2 training provided to stop smoking professionals, but tailored to the study and adding extra training on the use of the e-cigarette pod device specifically. Advisors were funded for the trial as an excess treatment cost, meaning that future commissioning of the intervention, if implemented, would include the justification of the need for a specific person to deliver stop smoking support (possibly combined with other lifestyle interventions in routine practice).

10. I think the behavioural elements of the intervention could be better specified. You mention it is theory driven what theory is this. Also has the intervention been feasibility tested. Reference to this and any related publications would be helpful.

A reference has been added to the 'Behaviour change wheel' demonstrating that the COM_B model of health behaviour change is the theory underpinning the behavioural component of the intervention. We will make the intervention manual fully open access on completion of the trial where much more detail is shared about this aspect of the intervention.

Primary outcome

I am a little unclear whether it is self-report that is primary outcome or CO reading. What will happen if self-report not support by CO reading?

The primary outcome is CO verified abstinence. If the self report is different, we go by the CO verification. If CO readings cannot be gathered, the participant is assumed to be smoking, by intention to treat.

References

These seem to be have gone a little awry from 17 onwards e.g. 19 is not the appropriate fidelity reference

These have all been checked and corrected

Table one

Check wording it currently says 'eligibility screening including' on third row but then doesn't include inclusion.

Apologies as the row had not been expanded. This has been addressed

VERSION 2 – REVIEW

REVIEWER	Steed, Liz Barts and The London School of Medicine and Dentistry, Centre for Primary Care and Public Health
REVIEW RETURNED	11-Nov-2022

GENERAL COMMENTS	This remains an important and largely well written protocol with most issues addressed but some important outstanding revisions include:-  1. In relation to adding a definition of 'unmotivated'. I think you mean to say NOT actively seeking smoking cessation rather than NOW actively seeking smoking cessation:- 2. References are still incorrect with reference to the NIH BCC still not added. It currently reads as 19 Michie but this is not the appropriate reference, the appropriate reference should be added and subsequent references updated. 3. The authors have kindly clarified to me as a reviewer that 'If CO readings cannot be gathered, the participant is assumed to be smoking, by intention to treat.' The point was to clarify this in the manuscript - such that regardless of self-report status, if non-CO verified they will be counted as smokers.
---

VERSION 2 – AUTHOR RESPONSE

We thank Dr Steed for her careful attention to our protocol paper. We have addressed all remaining comments, specifically:

1. In relation to adding a definition of 'unmotivated'. I think you mean to say NOT actively seeking smoking cessation rather than NOW actively seeking smoking cessation:-
Quite right! We had amended this in the strengths and limitations section but had not spotted this typo in the introduction. Thank you.

2. References are still incorrect with reference to the NIH BCC still not added. It currently reads as 19 Michie but this is not the appropriate reference, the appropriate reference should be added and subsequent references updated.

Again, thank you for spotting this. This was a reference from an old version of our manuscript that had escaped our reference editing tool. This has now been updated, as have all subsequent references.

3. The authors have kindly clarified to me as a reviewer that 'If CO readings cannot be gathered, the participant is assumed to be smoking, by intention to treat.' The point was to clarify this in the manuscript - such that regardless of self-report status, if non-CO verified they will be counted as smokers.

On the tracked changes version of the manuscript you will see that I have highlighted where this point was already clarified, in the analysis section. I had not added additional text to the primary outcome statement as we reference the Russel standard, where this approach to missed verification by ITT is part of the guideline. However, for absolute clarity, I have also now included additional clarification in the text, as requested.